## Research Article

PTSD; human rights; trauma-informed; Latin America; immigration

**Corresponding author:**
Alejandro Luis Vázquez;
Email: avazquez4004@gmail.com

# Trends in trauma: Increasing rates of sexual and domestic violence among female Latin American asylum seekers

Alfonso Mercado[1] , Andy Torres[1], Francisco Banda[1], Amanda Venta[2], Luz Garcini[3], Alejandro Luis Vázquez[4] and Aldo Barrita[5] and Oswaldo Moreno[6]

[1]The University of Texas Rio Grande Valley, USA; [2]University of Houston, USA; [3]Rice University, USA; [4]The University of Tennessee Knoxville, USA; [5]Michigan State University, USA and [6]Virginia Commonwealth University, USA

## Abstract

The last 10 years of scientific research analyzing asylum-seekers' mental health has established high rates of trauma exposure throughout the migratory trajectory. However, limited studies have identified gender-based violence among Central American asylum-seeking women. The purpose of this study was to identify the frequencies of gendered-base violence among asylum-seeking women from Central and South America at a humanitarian respite center (USA) and a tent encampment (Mexico) on both sides of the United States–Mexico Border using data from three independent studies in 2016, 2019, and 2023, respectively. Visual trend analysis identified a peak in domestic violence in 2019, a stable frequency of sexual assault across the three studies, and a downward trend in Study 3 compared to Study 1 for all types of gender-based violence except for domestic violence. Age stratification revealed diverse patterns in trauma rates. Trends in domestic violence differed between the 18–25 (56%) and 26+ years age groups (70%), in one study, substantially higher than the prevalence of the 29% rate among US female community samples. The data highlights the need for immigration reform addressing women's human rights and provides insights for mental health service providers to promote trauma-informed care for this vulnerable immigrant group.

## Resumen

Los últimos diez años de investigación científica sobre la salud mental de las personas solicitantes de asilo han establecido altas tasas de exposición a trauma y de trastorno de estrés postraumático (TEPT), así como de malestar relacionado con el trauma a lo largo de la trayectoria migratoria. Sin embargo, persiste una brecha en la identificación de tipos específicos de experiencias traumáticas entre las mujeres solicitantes de asilo. Tomando en cuenta los llamados internacionales para identificar y prevenir la agresión sexual y delitos relacionados contra personas solicitantes de asilo, el propósito de este estudio fue identificar las frecuencias de violencia sexual y violencia doméstica entre mujeres solicitantes de asilo provenientes de Centro y Sudamérica en dos centros humanitarios de acogida y un campamento de tiendas de campaña a ambos lados de la frontera entre Estados Unidos y México, utilizando datos de tres estudios independientes. Los datos fueron recopilados en tres estudios realizados en 2016 (Estudio 1), 2019 (Estudio 2) y 2023 (Estudio 3). Las tendencias visuales identificaron un pico en la violencia doméstica en 2019, una frecuencia relativamente estable de agresión sexual a lo largo de los tres estudios, y una tendencia a la baja en el Estudio 3 en comparación con el Estudio 1 para todos los tipos de violencia de género, excepto la violencia doméstica. La estratificación por edad reveló patrones diversos en las tasas de trauma. Las tendencias en violencia doméstica difirieron entre los grupos de 18 a 25 años (56%) y de 26 años o más (70%), en un estudio, siendo sustancialmente más altas que la prevalencia del 29% observada en muestras comunitarias femeninas en Estados Unidos. Estos datos destacan la necesidad de reformas migratorias que aborden los derechos humanos de las mujeres y proporcionan información clave para profesionales de la salud mental, con el fin de promover una atención informada en trauma en un contexto de aumento de delitos sexuales y violencia doméstica en este grupo inmigrante vulnerable.

## Impact Statement

This study provides critical evidence on gender-based violence among Latin American asylum-seeking women, with crucial implications for immigration and international policy, clinical practice and research directions. From a policy perspective, the high rates of gender-based violence, particularly among women aged 26 years and older, highlight the urgent need for immigration reform that incorporates gender-responsive protections, including improved screening for endured violence, expanded legal safeguards and access to safe reporting

mechanisms and shelters throughout the migratory process. Clinically, the findings highlight the necessity of trauma-informed, culturally responsive care that accounts for varied trauma exposures across migration stages, as well as age-specific patterns in victimization; mental health providers must be competent to deliver comprehensive trauma screening, early intervention and sustained support to address trauma-stressors and related culture-bound distress symptoms in this vulnerable population. From a research standpoint, these results reveal ongoing gaps in identifying unique and underreported forms of trauma among asylum-seeking women, pointing to the need for longitudinal and intersectional studies that examine how factors such as age, migration pathways. Additionally, future research should evaluate the effectiveness of policy and clinical interventions designed to reduce violence and improve mental health outcomes. This study strengthens the evidence base necessary to inform targeted policy reform, enhance clinical care, and guide future scholarship aimed at protecting the human rights and health of asylum-seeking women.

## Introduction

Global migration to the United States has increased exponentially since 2020, peaking at 2,475,669 US Border Patrol encounters in the Southwest land border in 2023 (United States Customs and Border Protection, 2024). Within this broader figure, 495,286 migrants come from the Northern Triangle region of Central America (United States Customs and Border Protection, 2024), a group of asylum seekers that endures exceptionally high levels of traumatic events before, during and after their migratory journey (Mercado and Venta, 2022). Current figures from the US Border Patrol reflect an increase in the influx of asylum seekers; for instance, estimates from 2019 indicate that ~470,000 people from the Northern Triangle sought asylum in various countries, citing domestic and community violence and sociopolitical turmoil (United Nations High Commissioner for Refugees, 2019) reflecting a fivefold increase when compared to the 2014 estimates (United Nations High Commissioner for Refugees, 2019).

Though trauma exposure and trauma-related stress are prevalent across the board in Central American asylum-seeking samples (Mercado et al., 2021; Mercado and Venta, 2022; Torres et al., 2022), there is growing concern about the vulnerability of Latina asylum seekers, given high rates of domestic and sexual violence reported in prior studies (Torres et al., 2022). Of particular interest are later symptoms that are sequelae of trauma events or potentially traumatic events that are not explained by the Western concept of post-traumatic stress; instead, asylum seekers from Central and Latin America appear to manifest internalizing, externalizing and affective and mood, and somatic symptoms in unique typologies related to culturally reinforced idioms or "shapes" of distress (Navarro Flores et al., 2023). Table 1 describes the common symptoms and diagnoses endorsed by asylum seekers from Latin America. Despite clinical concerns and anecdotal reports of increased incidence, few studies have focused on assessing violence perpetration among Latina asylum seekers specifically, and, to the best of our knowledge, nothing is known about how levels of exposure are changing across time (Torres et al., 2022; American Psychological Association, 2024). Within this context, this study aimed to describe the rates of reported domestic and sexual violence among asylum-seeking Latina immigrants in the United States from data collected in three separate studies spanning from 2017 to 2023.

### Asylum-seeking women and gender-based violence

Previous research finds that women and men cite significantly different reasons for fleeing their home nations (Castañeda et al., 2015). For example, women are more likely to experience gender-based violence across their pre-migratory, migratory and post-migratory journeys (Menjívar et al., 2019; Torres et al., 2022).

One study found that that 77% of a sample of Central American mothers fleeing to the United States cited "gang violence" and 33% cited "domestic abuse" as the top two reasons for fleeing (MacLean et al., 2019). Moreover, women are more likely than men to be exposed to human rights violations, such as family separations and forced maternity, among others (Castañeda et al., 2015). In this view, it is worth directing attention to feminist models of asylum-seeking women and mothers (Hernández, 2019).

Amid increasing trends of community and gang violence, impoverished environments, failed states, sociopolitical turmoil and recent waves of natural disasters in Central and South America (Barriga Cabanillas et al., 2014; Mercado and Venta, 2018, 2022; Menjivar et al., 2019; Menjívar and Walsh, 2019; Villa et al., 2021; Torres et al., 2022; Fitzgerald, 2023), women have been historically more affected by these socioecological factors than men. As seen by Bianco (2019), immigrant and asylum-seeking women and mothers already experience high risks for traumatic experiences and systemic inequalities in their home nations, in their migratory journey and even in the process of their acculturation in the United States (Bianco, 2019; Torres et al., 2022). As such, it is imperative to identify the current state of health and traumatic experiences of asylum-seeking women from the Northern Triangle to take steps toward alleviating these inequities and vulnerabilities to trauma.

### Gender-based violence for Latinas in Central and South America

As per the United Nations General Assembly, violence against women is defined as the "act of gender-based violence that results in, or is likely to result in physical, sexual, or psychological harm or suffering to women, including threats of such acts, coercion, or arbitrary deprivation of liberty, whether occurring in public or in private life" (p. 2) (United Nations General Assembly, 1993). In the current study, we focus on two forms of violence against women: *domestic* and *sexual* violence that is either directly experienced or witnessed.

In a CDC report by Bott et al. (2014) sampling prevalences of gender-based violence in Latin America, lifetime prevalence rates of physical violence (PV) by an intimate partner (IP) ranged from 13.4% (Haiti) to 52.3% (Bolivia). In the same study, the lifetime prevalence of intimate partner physical violence was 24.5% for Guatemala, 24.2% for El Salvador and 38.6% for Colombia. The Bott et al. (2014) report identifies data from other countries in Central and South America that were not sampled in this study. It is difficult to draw comparisons and conduct inferential statistics on these data, given that each country used different sampling methods, questions to ascertain violence and dissemination procedures (Bott et al., 2014). Nonetheless, the data support the notion that asylum-seeking women from the region are already exposed to gender-based

**Table 1.** Conditions endorsed by Latin American asylum seekers

| Name of conditions endorsed by asylum seekers | DSM-5TR symptoms or medical criteria | Examples of endorsements |
|---|---|---|
| Posttraumatic Stress Disorder (APA, 2024) | *Trauma- and stress-related disorder caused by direct exposure or witnessing a death, injury or violence (sexual or physical).*<br>*Criterion includes:*<br>*A: Direct or witnessed exposure*<br>*B: Intrusion symptoms (flashbacks, nightmares and intrusive memories about the event)*<br>*C: Avoidance (active avoidance of the event, including people, places and thoughts)*<br>*D: Changes in cognition and mood (negative changes in mood, inability to remember key aspects of the stressful event)*<br>*E: Arousal and reactivity (hypervigilance, irritability and angry outbursts)* | Flashbacks about family separation<br>Refusal to talk about family separation<br>Irritable mood or "acting out" expressed as anger at "the system"<br>Intrusive memories or thoughts about the traumatic journey or about witnessing deaths of fellow friends or family during the journey<br>(Mercado and Venta, 2022) |
| Depression (APA, 2024) | Depressed mood and anhedonia for at least 1 month and at least 3–4 or more of the following:<br>Weight/appetite change<br>Sleep disturbance<br>Psychomotor agitation/retardation<br>Fatigue<br>Worthlessness/guilt<br>Inability to concentrate<br>Indecisiveness<br>Suicidal ideation | Loss of interest, joy and energy to complete daily activities<br>Feeling guilty about the traumatic events, family separation<br>Inability to concentrate or complete their daily tasks, like their immigration paperwork<br>(Mercado and Venta, 2022) |
| *Ataque de Nervios* (Guarnaccia et al., 2010) | No diagnostic criteria established by the American Psychiatric Association; the *Ataque de Nervios* is typically used as a quantitative measure<br>1. *Shouted a lot*<br>2. *Had crying attacks*<br>3. *Broke things or became aggressive*<br>4. *Became very angry or enraged*<br>5. *Felt very scared or frightened*<br>6. *Became hysterical*<br>7. *Trembled a lot*<br>8. *Felt strange, as if it was not you this was happening to*<br>9. *Had a period of amnesia*<br>10. *Felt dizzy*<br>11. *Fell to the floor with convulsions (a "seizure")*<br>12. *Had heart palpitations (heart beating hard)*<br>13. *Felt chest tightness or heat in the chest*<br>14. *Fainted or felt on the verge of fainting*<br>15. *Tried to hurt yourself or attempted suicide* | Feeling very scared about the immigration process<br>Trembling/shaking when seeing news about immigration processes<br>Fainting-like or feelings as if they are about to faint when thinking about the immigration process |
| Somatization/Pain Complaints (APA, 2024; Garcini et al., 2022; Lewis-Fernandez, 2016). | Pins and needles in the hands or feet<br>Heat sensations inside the body | Having pain in arms and legs after seeing news about fellow nationals crossing the migration points in the Panama jungle "*El Darien*" |

violence and are likely to experience it across various points in their migratory trajectory.

### Sexual violence

While it is difficult to find recent and transparent statistics of gender-based violence from national sources, existing research indicates that asylum-seeking Latinas from Central America experience higher levels of domestic violence in their home nations compared to men (Menjivar et al., 2019; Menjívar and Walsh, 2019). Sexual violence against women can occur in various migratory points, even in locations with the relative protection of local authorities, such as respite centers, temporary stops or transition points and monetary resource sites (*e.g.*, money exchange services) (Soria-Escalante et al., 2022). Immigrant women and mothers are also at risk for sexual exploitation, given their often high need for sustenance, medication or access to shelter in an unpredictable

migratory journey (Mercado and Venta, 2022; Soria-Escalante et al., 2022; Torres et al., 2022).

Instances of sexual assault can also be lived vicariously, as these instances are often shared among women migrating together (Soria-Escalante et al., 2022). A report from the World Bank estimates that about 40% of Central American women seeking asylum in the United States have witnessed a form of violence, including physical, sexual and psychological violence. Although Central American asylum seekers are already a vulnerable population for community violence, gun violence and other trauma/potentially traumatic events, the female subpopulation is particularly at higher risk for sexual assault (Menjivar et al., 2019; Menjívar and Walsh, 2019; Mercado et al., 2022; Torres et al., 2022).

In the Bott et al. (2014) report, the lifetime prevalence rates of sexual violence by a partner ranged from 5.2% (Dominican Republic) to 15.2% (Bolivia). The lifetime prevalence was 12.3% for

Guatemala, 11.8% for Colombia, 11.5% for El Salvador and the 12-month prevalence for Honduras was 5% (Bott et al., 2014). The rates become difficult to compare differences in item choices in the surveys for all the countries reporting the data.

In Speizer et al. (2008), 12.6% of asylum-seeking women from El Salvador endorsed sexual abuse at any age, and 6.4% experienced sexual abuse before the age of 15 years. The same study sampled 8,860 Guatemalan women seeking asylum and identified a 7.1% prevalence rate of sexual abuse at any age, with 6.4% of women experiencing sexual abuse at any age before the age of 15 years (Speizer et al., 2008). These data points once more cannot be compared due to various methodological differences in data collection and time range (*e.g.*, lifetime data and annual rates). Nonetheless, they present a snapshot of the high rates of gender-based violence reported by this immigrant group.

### Domestic violence

Central American asylum seekers cite intimate partner violence and threats as the two main reasons for seeking asylum in the United States (Menjivar et al., 2019). A different study reported that 33% of asylum-seeking mothers from Central and South America endorse domestic violence as a reason for fleeing their home nation (MacLean et al., 2019). Specifically, gender-based violence perpetrated by the intimate male partner, extortion and gang violence consistently appeared as primary reasons for Central American asylum seekers to seek asylum in the United States (Menjivar et al., 2019; Menjívar and Walsh, 2019). The data identified a 24.5% prevalence of women reporting any instance of physical violence by a partner in Guatemala during the 2008–2009 fiscal year; the 2008 rate for El Salvador was 24.2% (Bott et al., 2014), and there were no lifetime data for women from Honduras (Bott et al., 2014).

In a sample of women asylum seekers from Honduras, El Salvador and Guatemala, participants endorsed a total of 21%, 13% and 10% of intimate partner violence, respectively (Baranowski et al., 2020). Specifically, the participants stated that they witnessed or lived domestic violence, which had themes of "power and control," and often saw the abusive behavior as normal in the context of their home countries and societal expectations (Baranowski et al., 2020). The limited data available carries implications for the sociocultural risk factors experienced by this immigrant group in the context of domestic violence.

### The current study

The purpose of this study was to identify the frequencies of sexual violence and domestic violence among asylum-seeking women from Central and South America at two humanitarian respite centers and a tent encampment on both sides of the United States–Mexico Border using data from three independent studies.

### Methods

### Participants

#### Study 1

This study had a total of 103 participants. The study had a mean age of 32.21 (standard deviation [SD] = 7.58). Most of the sample self-identified as female (55.34%), followed by those who self-identified as male (41.75%). In this sample, most participants were from Honduras (43.75%) and El Salvador (41.7%).

#### Study 2

This study had a total of $n = 51$ participants. Most of the sample self-identified as female (70.6%). The sample had a mean age of 51 (SD = 28.02). Most of the sample was from Honduras (62.7%) and Guatemala (25.5%).

#### Study 3

Study three had a total of 422 participants. The sample had a mean age of 31.40 (SD = 9.12). The majority of participants were female (55.45%). Most of the sample was from Honduras (43%), followed by those from Venezuela (19%).

### Procedures

#### Design

Table 2 describes the methodology, including study location, data collection period, immigration policy context amid the collection period, sample size and basic demographics, violence measures and key differences from each study when compared to other studies.

**Study 1** is part of an international study by Mercado and Venta (2022). The data were collected during 2016–2017 at a humanitarian respite center at the Texas–Mexico border. This crossroad is for asylum seekers from Mexico, Central America, South America and other international migration points (Venta and Mercado, 2019). At this respite center, families arrive after being processed by US Customs and Border Patrol. They are provided with basic human necessities (*e.g.*, food, water, shelter and emergency needs). Families typically stay at this shelter for 24 h and receive aid for their continued trip within the United States.

The lead author's [RETRACTED] institution and Institutional Review Board (IRB) approved this study. During periods when families were resting and waiting for their departure to their next destination in the United States (typically to their sponsoring family member), they were approached by a team of bilingual researchers. The team included US Bilingual Psychologists and supervised graduate students. All consent for this study was provided in Spanish. Researchers invited families to participate in this study, explaining that it was solely for research and academic purposes and would not affect their migration or legal processes. Families were given a $20 USD gift card upon completion of the study. Families agreeing to participate were interviewed in Spanish using a demographic questionnaire and psychological measures.

**Study 2** is part of a broader international study by American Psychological Association (2024). This study was very similar to *Study 1*. The data were collected during 2019–2021 at the same humanitarian respite center at the Texas–Mexico border with families affected by Migrant Protection Protocol policies. Once more, the bilingual researchers approached the families when they were resting and waiting for their departure to their next destination. This was the same team composed of US Bilingual Psychologists and supervised graduate students. Researchers invited families to participate in this study, explaining that it was solely for research and academic purposes and would not affect their migration or legal processes. Families were told to receive a $30 USD gift card upon completion of the study. Families agreeing to participate were interviewed in Spanish with a demographic questionnaire and psychological measures. IRB approval was granted for this study. All consent for this study was provided in Spanish. Further information on the logistics of this study is presented in other studies (Mercado et al., 2022; Morales et al., 2023).

**Study 3** is part of a much larger longitudinal NIH-funded study by the authors Venta et al. (2024). It included a total of

**Table 2.** Methodologies and logistics per study

| Study # | Data collection period | Location (US–Mexico Border Context) | Immigration policy context | N | Participant age | Violent measures (DV/SV) | Key difference from other studies |
|---|---|---|---|---|---|---|---|
| #1 | 2016–2017 | Humanitarian respite center, Texas–Mexico border | Pre-MPP | 57 | 29.56 (M = 29.56, SD = 6.84) | HTQ items: HTQ–7 Sexual assault; HTQ–8 Other sexual violence; HTQ–17 Witnessed sexual assault; HTQ–18 Domestic violence | Smallest sample; first data collection point; IRB approved; temporal synthesis approach; not pooled statistical comparison; bilingual teams; informed consent in Spanish. |
| #2 | 2019–2021 | Same humanitarian respite center, Texas–Mexico border | During MPP | 35 | 28.09 (M = 28.09, SD = 7.38). | THQ-SF items: THQ–18 Sexual assault; THQ–19 Other sexual harassment; THQ–23 Domestic violence | Focused on MPP-affected families; older mean age; different instrument from Studies 1 and 3; IRB approved; temporal synthesis approach; not pooled statistical comparison; bilingual teams; informed consent in Spanish; participants received a gift card; THQ does not capture *indirect* exposure |
| #3 | 2023 | Two locations: McAllen Humanitarian Respite Center (TX), Senda de Vida tent encampment (Reynosa, Mexico) | Post-MPP; NIH-funded longitudinal | 213 | (M = 30.19, SD = 8.48). | HTQ items: HTQ–7 Sexual assault; HTQ–8 Other sexual violence; HTQ–17 Witnessed sexual assault; HTQ–18 Domestic violence | Largest sample; multi-site collection including Mexico; longitudinal focus; IRB approved; temporal synthesis approach; not pooled statistical comparison; bilingual teams; informed consent in Spanish; participants received a gift card |

406 participants. The data were collected during 2023 at two locations. The first location was the same humanitarian respite center at the Texas–Mexico border, along the Rio Grande Valley. The second location was at *Senda de Vida*, a tent encampment in Reynosa, Tamaulipas, Mexico. This is another common stop for asylum seekers from Central America and South America as they await their asylum hearing request to the US Tent encampments in Mexico provided shelter and basic human necessities to families arriving at the United States–Texas border. Both locations for this study (and by *de facto* of all studies) are presented in Figure 1. The McAllen Humanitarian Respite Center, located in South Texas, is ~30 min (by vehicle) from the Texas–Mexico border. In comparison, the *Senda de Vida* center is ~15 min, by transport from the Mexico–Texas entry point. The team of researchers was divided into two:

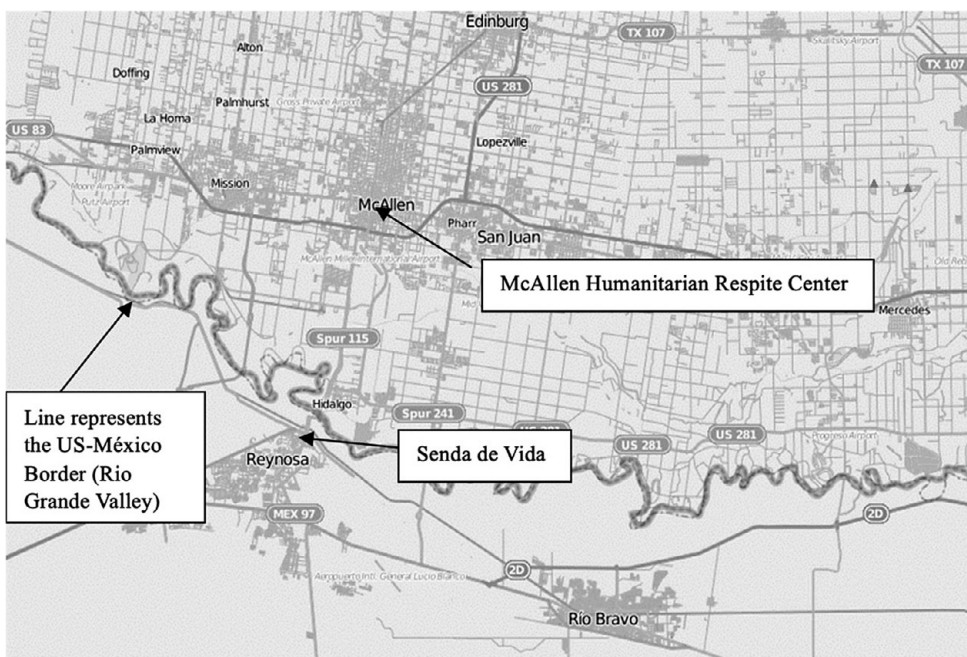

**Figure 1.** Reynosa–McAllen map with study sites and border annotated. Map by OpenStreetMap contributors, modified by Andy Torres, CC BY-SA 2.0, via Wikipedia.

Team 1 attended the US data collection (McAllen Humanitarian Respite Center), and Team 2 attended the Mexico data collection point (*Senda de Vida*).

During 2019, the United States enacted migration protocols (commonly referred to as "*Remain in Mexico*" or the Migrant Protection Protocol [MPP]) that denied asylum seekers entrance to the United States, forcing these immigrant groups to exit the country and resume or start their asylum protocol in Mexico, contriving a situation of logistical and psychological distress (Mercado et al., 2021, 2022).

Author RETRACTED's institution IRB approved of this study. All consent for this study was provided in Spanish. A team of bilingual researchers interviewed families in Spanish and collected basic biometric data. Researchers explained that it was solely for research and academic purposes and would not affect their migration or legal processes. Families were told they would receive a $30 USD gift card upon study completion. Families agreeing to participate were interviewed in Spanish.

### Measures

#### Traumatic experiences
**Study 1 and study 3.** To measure the number of traumatic experiences, the authors deployed the Spanish version of the *Harvard Trauma Questionnaire* (Mollica et al., 1992). This scale includes a 24-item checklist that asks families whether they have experienced traumatic events. Participants were asked to indicate "*No*" or "*Yes*" to each item. Only the gender-based trauma items were extracted for all formal analyses, namely the *HTQ-7* (Sexual assault), *HTQ-8* (other types of sexual violence, *e.g.*, sexual harassment), *HTQ-17* (*e.g.*, witnessing sexual assault) and *HTQ-18* (domestic violence: physical, verbal or emotional). This scale was included as part of the measures in Study 1 and Study 3.

**Study 2.** This study used a scale different from that of *Study 1* and *Study 3*. Study 2 used the Spanish version of the Trauma History Questionnaire–Short Form (Hooper et al., 2011), a 24-item checklist that asks parents whether they have experienced 24 possible traumatic events. Examples of the items include "*Has anyone ever tried to take something directly from you by using force or the threat of force, such as stick-up or mugging?*" Participants were asked to indicate "*No*" or "*Yes*" to each item and subsequently to indicate the

"Number of times" the instance occurred and the "Approximate age(s)" of the instance(s). For this study, the authors selected items related to gender-based violence, namely the THQ-18 (sexual assault), HTQ-19–Short Form (other forms of sexual harassment) and THQ-23 (domestic violence). The THQ does not identify *indirect* exposure to sexual assault.

### Results

#### Demographics

*Study 1* had a total of 57 participants with a mean age of 29.56 ($M = 29.56$, SD = 6.84). *Study 2* had a total of 35 participants and had a mean age of 28.09 ($M = 28.09$, SD = 7.38). *Study 3* had a total of 213 participants and had a mean age of 30.19 ($M = 30.19$, SD = 8.48). Nationalities for each sample are listed in Table 3.

#### Sexual violence rates
For *Study 1*, a total of 21.8% of the sample endorsed sexual assault, and a total of 26.3% endorsed "other form of sexual assault." *Study 1* did not include a survey item on "Witnessing sexual assault." For study 2, the sample endorsed 2.9% of sexual assault, 5.7% of "Other form of sexual assault" and 47.5% of witnessing sexual assault. For *Study 3*, the sample endorsed a total of 14.1% prevalence of sexual assault, 19.7% of "Other form of sexual assault" and 9.95% of witnessing sexual assault. These are reported in Table 4.

#### Domestic violence rates
*Study 1* had a total of 21.8% of the sample who endorsed domestic violence. *Study 2* had a total of 62.9% of the sample who endorsed domestic violence. *Study 3* had a total of 38% of the sample who endorsed domestic violence. Table 4 describes the rates of traumatic experiences reported by participants.

#### Trends per study
When compared to *Study 1*, all categories of gender-based violence showcased a downward visual trend in *Study 3*, except for "domestic violence." Study 1 had similar rates of sexual assault, Other sexual assault and domestic violence rates. In Study 2, the most endorsed rate was DV, followed by witnessing DV. Study 2 also had the lowest rates of sexual assault or "other forms of sexual assault"

**Table 3.** Demographic cross-tabulation of country by study number among asylum-seeking Latin American women

| Variable | Age by country | | Study 1 (*n* = 57) | | Study 2 (*n* = 35) | | Study 3 (*n* = 213) | |
| | *M* | SD | *n* | % | *n* | % | *n* | % |
| --- | --- | --- | --- | --- | --- | --- | --- | --- |
| Age by study Sample (*M*, SD) | | | 30.87 | 7.70 | 28.09 | 7.38 | 30.19 | 8.48 |
| Country | | | | | | | | |
| *Venezuela* | 29.56 | 6.84 | n/a | n/a | n/a | n/a | 50 | 23.5 |
| *Mexico* | 32.47 | 10.70 | 2 | 3.50 | n/a | n/a | 15 | 7.00 |
| *Honduras* | 29.52 | 8.46 | 28 | 49.10 | 24 | 68.60 | 99 | 46.50 |
| *Guatemala* | 28.94 | 6.03 | 6 | 10.50 | 7 | 20.00 | 19 | 8.90 |
| *El Salvador* | 31.36 | 7.98 | 21 | 36.80 | 4 | 11.40 | 9 | 4.20 |
| *Colombia* | 32.43 | 10 | n/a | n/a | n/a | n/a | 21 | 9.90 |

*Note*: One-way analysis of variance showed no significant differences in mean age across countries; $F (5, 300) = 1.06$, $p = .38$. One-way analysis of variance showed no significant differences in mean age across studies; $F (2, 300) = 1.32$, $p = .27$.

**Table 4.** Rates of gender-based traumatic experiences in asylum-seeking females from Latin America

| Traumatic experience | Rates of trauma experiences |
|---|---|
| **Sexual assault** | |
| *Study 1* | 12 (21.1%) |
| *Study 2* | 1 (2.9%) |
| *Study 3* | 30 (14.1%) |
| **Other forms of sexual assault** | |
| *Study 1* | 15 (26.3%) |
| *Study 2* | 2 (5.7%) |
| *Study 3* | 42 (19.7%) |
| **Domestic violence** | |
| *Study 1* | 12 (21.8%) |
| *Study 2* | 22 (62.9%) |
| *Study 3* | 78 (38%) |
| **Witnessing sexual assault** | |
| *Study 1* | 8 (14%) |
| *Study 2* | 16 (45.7%) |
| *Study 3* | 21 (9.95%) |

and the highest difference in rates of trauma within the study. Finally, in Study 3, the most endorsed form of gender-based trauma was DV, followed by "other form of sexual assault." Excluding DV, Study 3 had relatively identical rates of "other form of sexual assault," "sexual assault" and "witnessing DV."

The studies were correlated with three major worldwide events, including asylum policy changes for Central America (Study 1) (Mercado and Venta, 2022), the global pandemic of COVID-19 and MPP policies (Study 2) and the aftereffects of MPP policies (Study 3). Thus, each study and its corresponding worldwide events in the United States–Mexico border likely created unique sociopolitical factors that contrived distinct situations that likely influenced the responses of participants.

A visual analysis of the studies is presented in Figure 2, noting a change in rates of sexual assault, other forms of sexual assault and domestic violence. As a supplementary analysis, the author of this study also conducted a trend analysis by further stratifying it by age group. Table 4 of this study shows the frequencies of traumatic injuries stratified by study wave and age group.

As visualized in Figure 3 and Table 5, age stratification revealed concurrence between the two age groups in all trauma rate trends, with the sole exception of sexual assault. For the 18–25 age group, sexual assault rates demonstrate a downward trend. Meanwhile, sexual assault rates in the 26+ age group show a U-shape such that study two shows a drop to 0% before returning in study three back to about an equal level as study one.

## Discussion

Central and South American communities have been historically impacted by community violence, extreme poverty, natural disasters and limited health and financial infrastructures, which hinder the promotion of optimal mental health (Cardeli et al., 2022; Mercado and Venta, 2022; Torres et al., 2022; American Psychological Association, 2024). Additionally, gender-based violence has been cited as one of the main reasons for Latin American women fleeing their home countries and embarking on their migratory journey to the United States (Mercado and Venta, 2022; American Psychological Association, 2024). This study aimed to describe and disseminate the rates of four types of gender-based violence among asylum-seeking women in humanitarian respite centers on both sides of the United States–Mexico border. The findings from this study provided an overview of the frequency of observed changes in four types of gender-based violence in asylum-seeking women from Latin American countries.

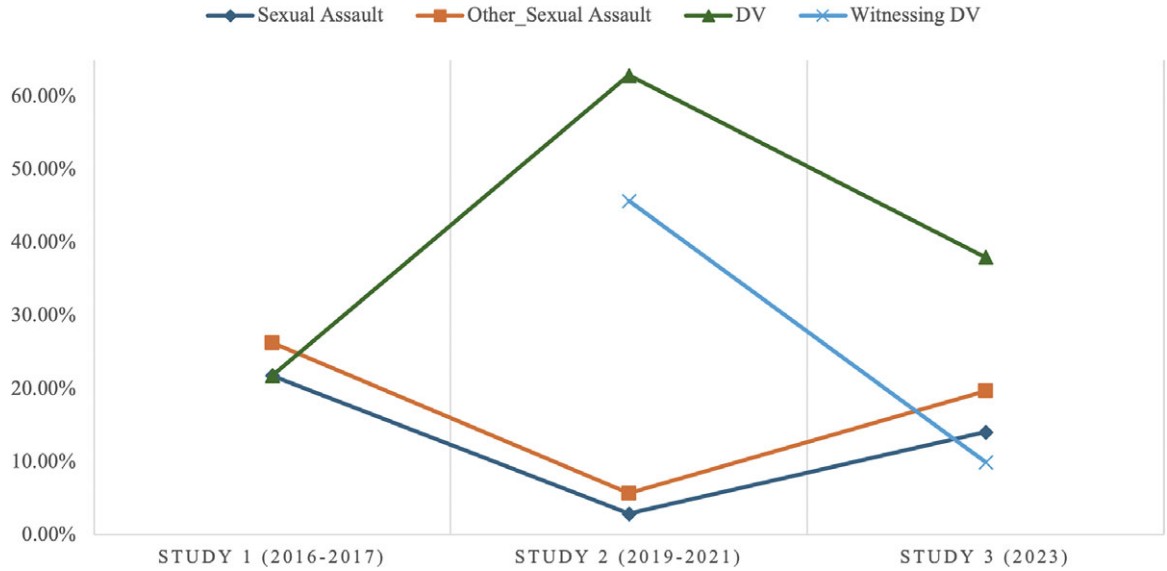

**Figure 2.** Line graph illustrating trends of gender-based traumas across the three studies.

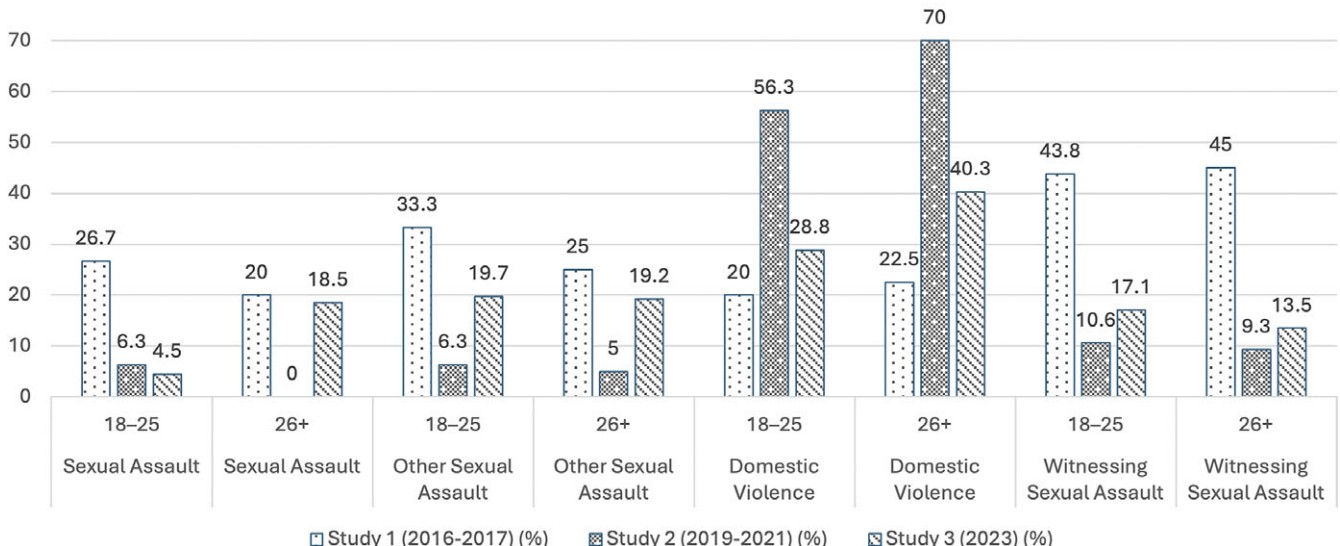

**Figure 3.** Bar graph visualizing rates of gender-based traumas across the three studies as stratified by age.

**Table 5.** Rates of gender-based traumatic experiences in asylum-seeking women from Latin America, by age group and study number

| Traumatic experience | 18–25 Years (n = 97) | Older than 25 years (n = 214) |
|---|---|---|
| *Sexual assault* | Study 1: 4 (26.7%) | Study 1: 8 (20%) |
| | Study 2: 1 (6.3%) | Study 2: 0 (0%) |
| | Study 3: 3 (4.5%) | Study 3: 28 (18.5%) |
| *Other forms of sexual assault* | Study 1: 5 (33.3%) | Study 1: 10 (25%) |
| | Study 2: 1 (6.3%) | Study 2: 1 (5%) |
| | Study 3: 13 (19.7%) | Study 3: 29 (19.2%) |
| *Domestic violence* | Study 1: 3 (20%) | Study 1: 9 (22.5%) |
| | Study 2: 9 (56.3%) | Study 2: 14 (70%) |
| | Study 3: 19 (28.8%) | Study 3: 58 (40.3%) |
| *Witnessing sexual assault* | Study 1: 7 (43.8%) | Study 1: 9 (45%) |
| | Study 2: 7 (10.6%) | Study 2: 14 (9.3%) |
| | Study 3: 14 (17.1%) | Study 3: 23 (13.5%) |

## Visual trends

### Sexual assault

Study 1 had the highest rates of sexual assault across the three studies (21.8%). Study 2 had the lowest rates of "other forms of sexual assault" (5.7%) and "sexual assault" (5.9%). These frequencies remained relatively stable across the three independent studies, especially when considering the frequency fluctuations for the other traumatic experiences.

The National Sexual Violence Resource Center's (NSVRC) latest update, referencing 2015 data (Smith et al., 2018), estimates a 20% prevalence rate of sexual assault reported by women in the United States (National Sexual Violence Resource Center, 2015). Thus, the

highest reported rate of sexual assault across studies, 21.8% at Study 1, was equivalent to that observed in the United States. The data from Study 3 revealed a 14.1% rate of sexual assault, which is lower compared to US national statistics from the NIPSVS. It is possible that certain geopolitical and social events, like the COVID-19 pandemic, were at least partially responsible for the surge in sexual violence during Study 2. However, this remains difficult to map to the timeline of studies.

Another explanation for these findings is the likely impact of Migrant Protection Protocols (MPP) (Mercado et al., 2021). It has been argued that these policies worsened the mental health of Central and South American asylum seekers, given the increased psychological and logistical burden they imposed on the asylum-seeking process (Mercado et al., 2021). This immigrant group is likely to be more exposed to traumatic experiences, including organized crime, human rights violations and even natural disasters on the Mexican side of the US–Mexico border due to the Mexican government's limited resources (Mercado et al., 2021). Thus, the observed higher rates of sexual trauma among asylum-seekers may be at least partially explained by MPP (Mercado et al., 2021; Mercado and Venta, 2022). However, more data and analyses are needed to explore mechanisms for the observed correlation.

### Domestic violence

There was a peak in domestic violence (62.9%) experienced by the asylum-seeking women in Study 2, far exceeding the US prevalence rate of domestic violence among women of 29% in the United States, as per Black et al. (2011) cited by the US National Domestic Violence Hotline (n.d.) and of 41% of women as per Leemis et al. (2022) cited by the Centers for Disease Control and Prevention (CDC) (CDC, 2025). This comparison highlights the rather high rates of domestic violence among the cross-study findings of asylum-seeking women from Central and South America. This finding is expected as domestic violence and gender-based violence continue to be a commonly cited reason for seeking asylum in the

United States among many Latin American immigrant women. This point was the most endorsed form of traumatic experience across all three studies. Study 2 also had the highest rates of "witnessing DV" (47.5%) while having the lowest rates of sexual assaults. These rates reflect those from other studies (MacLean et al., 2019). The stories of many asylum-seeking families fleeing domestic violence do not begin in the United States but in their home nation or in their migratory journey (Alberto and Chilton, 2019). Further clinical and advocacy implications warrant evidence-based practices to protect asylum-seekers exposed to domestic violence.

## Clinical implications

Among asylum seekers, sexual assault has been well-established to be correlated with PTSD, depression, somatization and impaired adaptive function, among other psychological symptoms (Venta and Mercado, 2019; American Psychological Association, 2024). Mental health practitioners and medical providers can benefit from adapting a trauma-informed care practice using evidence-based practices to screen, assess, treat and refer clients/patients who are asylum seekers from Central and South America. The trauma-informed care practice should also rely on psychoeducation regarding how to navigate health and advocacy resources for asylum seekers. For instance, as per recommendations from primary care physicians, trauma-informed care with asylum seekers should maximize access to all available resources for this already vulnerable population; the resources should include but not be limited to advocacy centers, legal aid services, *pro-bono* mental health centers or university centers, food shelters, domestic violence shelters and so forth (Im and Swan, 2020). Working in low-income and underserved communities brings challenges when aiming to serve immigrant populations (Mercado and Venta, 2022; Palomin et al., 2023). Nonetheless, they also provide opportunities for leadership initiatives from licensed psychologists in mental health trauma-informed services. Psychologists should aim to bridge immigrants to advocacy centers, shelters and other social servicing agencies (Mercado and Venta, 2022).

Psychologists working with these populations should be aware of the recommended guidelines when conducting psychological evaluations used in immigration proceedings (Mercado et al., 2022). In both the evaluation and clinical intervention context, mental health providers should aspire to understand the complex effect of the migratory journey involving multiple instances of trauma across the entire migratory journey (Garcini et al., 2022; Mercado et al., 2022, American Psychological Association, 2024; Mercado and Venta, 2022). As many families face a critical need for psychological services, psychologists should become culturally competent to navigate this complex topic of immigration and trauma.

## Limitations

This study has various limitations. First, the study included a convenience sample, hindering generalization to other asylum-seeking families with different migratory trajectories. This study could also not compute inferential statistics due to the interdependence assumptions and differences in the instruments used. Future research could benefit from longitudinal designs to identify trends across traumatic experiences among asylum seekers to compute inferential statistics.

## Conclusion

In 2024, the American Psychological Association's Presidential Task Force on Immigration and Health released a timely and critical report (American Psychological Association, 2024) providing an overview of the changes in the immigration landscape. Focusing on the mesosystem of the ecological model of care (d'Abreu et al., 2019; Scharpf et al., 2021), the report makes several policy recommendations, including increased funding for immigrant well-being initiatives and the cessation of harmful practices, such as detention, separation and deportation (American Psychological Association, 2024). The findings of this current study align with these recommendations. This study underlines the need for psychologists to take on new advocacy roles to ensure the well-being of immigrants while buffering the impact of traumatic experiences.

**Open peer review.** To view the open peer review materials for this article, please visit http://doi.org/10.1017/gmh.2026.10212.

**Data availability statement.** Data are not available for this manuscript.

**Author contribution.** A.M.: Conceptualization, methodology, theoretical framework and writing–original draft preparation. A.T.: Methodology, conceptual analysis, data collection and writing–original draft preparation. F.B.: Data curation, investigation, critical review of the manuscript and data collection. A.V.: Supervision, methodology, and writing–review and editing. L.G.: Methodology, interpretation of findings and critical revision. A.V.: Writing–review and editing. A.B.: Writing–review and editing. O.M.: Writing–review and editing. All authors contributed meaningfully to the conceptualization and development of this manuscript. They were involved in the study design, interpretation of findings and critical revision of the content. All authors reviewed and approved the final version of the manuscript and agreed to be accountable for all aspects of the work.

**Financial support.** This work was supported by multiple funding sources. Primary support was provided by the National Institute on Minority Health and Health Disparities (NIMHD; Grant Nos. 1R01MD016897-01A1 and 1F31MD019521-01A1). Additional support was provided by the National Institutes of Health, National Heart, Lung and Blood Institute (NHLBI; Grant Nos. R56-HL174589 and 1R01HL174589-01A1), the Southwest Key Programs through the Substance Abuse and Mental Health Services Administration (SAMHSA) and internal seed funding from The University of Texas Rio Grande Valley. The content is solely the responsibility of the authors and does not necessarily represent the official views of the funding agencies.

**Competing interest.** The authors declare none.

**Ethics statement.** All procedures involving human participants were reviewed and approved by the appropriate Institutional Review Boards (IRBs; Protocol Nos. CR00003481 and IRB-19-0566). All participants provided informed consent in accordance with institutional guidelines and the ethical standards outlined in the Declaration of Helsinki.

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
