## [Reviewer Report]

I would like to thank the authors for their substantial investment and for the relevance of their work on this important and timely topic. The manuscript addresses a major public health and human rights issue, and the use of original, previously unpublished data collected across different periods is a clear strength.

The comments below are intended solely to improve the clarity and coherence of the manuscript and to help the reader more easily extract the key messages and added value of the study, thereby increasing its overall impact.

1. Major comment – Positioning of the study and justification of the analytical approach

The manuscript draws on data from three studies that have already been published, either as primary empirical work or as part of broader publications. While this is entirely legitimate, the rationale for re-analysing and jointly presenting these datasets would benefit from being made more explicit.

From the reader’s perspective, it is not entirely clear why the chosen approach was to present a new manuscript based on previously published studies, rather than:

Conducting a narrative or systematic review of the existing literature, or

Performing a formal meta-analysis if sufficient comparable data are available.

Without this clarification, a reader unfamiliar with the authors’ broader research programme might perceive the manuscript as a repackaging of previously published results rather than as a distinct and conceptually driven synthesis. Making the methodological and conceptual added value explicit would help prevent this interpretation and strengthen the positioning of the paper.

2. Major comment – Structure of results and synthesis of findings

Related to the point above, the current presentation of the Results section follows a study-by-study structure, which reinforces the impression of “three studies in one paper” and makes the overall message harder to extract.

Given that the individual studies are already available in the literature, the manuscript would benefit from a more synthetic analytical framework. Rather than detailing each study separately, the authors could focus on:

What the combined analysis of these datasets reveals,

The key overarching trends across time,

What new insights emerge specifically from considering these studies together.

A more integrated presentation would reduce redundancy, improve readability, and better highlight the contribution of this manuscript as a higher-level synthesis rather than a repetition of individual results.

3. Major comment – Introduction length and methodological coherence

The Introduction is comprehensive and well documented, but it is relatively long and would benefit from a clearer logical progression toward the study objective and design.

While the importance of studying violence among Latin American asylum seekers is well established, the introduction does not sufficiently explain:

Why a re-analysis and joint presentation of these previously published datasets is warranted,

Why this approach adds knowledge beyond what has already been published in the original studies,

Why this synthesis is particularly relevant at this point in time.

Strengthening this methodological reasoning would improve coherence between the background, the analytical choices, and the stated objectives, and would help the reader clearly understand the originality of the present work.

4

The Discussion section is rich and well informed, but at times quite broad, with extensive contextual and policy-related considerations. Its impact could be strengthened by more explicitly highlighting what this combined analysis adds beyond the original publications, and by clarifying what the reader should primarily retain from this temporal synthesis. Emphasising the specific contribution of the integrated approach would help distinguish the manuscript as a higher-level, integrative piece of work rather than as a juxtaposition of earlier studies. In addition, some minor adjustments could further improve overall readability, such as streamlining sections that are particularly dense in numerical results, limiting unnecessary decimal precision, and considering the relocation of certain detailed tables or descriptive elements to supplementary materials. These changes would not affect the substance of the work but would clearly enhance the reading experience.

Overall, this manuscript addresses a critical issue and builds on a solid body of prior research. I believe that clarifying the rationale for re-analysing previously published data, strengthening the synthetic dimension of the results, and tightening the methodological argumentation in the introduction would substantially enhance clarity, coherence, and impact. I would be happy to provide a more detailed and in-depth review once these points have been addressed, particularly as the current presentation—especially the tables, which combine multiple studies and contain a large number of numerical values—can be difficult to read. Streamlining the tables and limiting numerical precision (for example, rounding to a single decimal place) would further improve readability.

---

## [Reviewer Report]

Overall Comments to Authors:

This manuscript is overall very informative, this provides an interesting and needed examination of the needs of female asylum seekers in a critical sociopolitical climate. My major comments are mostly around the need to restructure the introduction, discussion, conclusion around the study population and how this work can advance migrant women’s health.

Abstract

Rather than including so much background on the topic, I would encourage authors to summarize the 3 studies a bit more (where they took place i.e. respite center on the border and a total N across all 3 studies, and the study design i.e. longitudinal/cross-sectional).

To me, visual trends sounds a bit off to say in an abstract so I would say “trends”

Introduction:

Overall this is well thought out but I think the authors should work on restructuring and really focusing on just the needs of Latina asylum-seekers. All the details on migration and gender-based violence and differences with men vs women are not totally necessary. Instead, I would go into more detail around conceptual framing (feminist theories you mentioned you used (p.4 line 36) and demonstrating the burden of gender based violence on their mental health (i.e., this only comes up in clinical implications section at the end of the paper p.19 lines 49-54).

Some more specific feedback:

p. 3 line 10- 495,286 may be better presented as a percentage

p.3 line 31—can you just reference once rather than twice in the same paragraph since it is the same source

p.3 line 33- “across the board” is very colloquial, I would delete this

p.3 line 42—can you describe what these symptoms are? You do this in clinical implications section at the end of the paper p.19 lines 49-54) and I think this needs to be in the introduction.

p. 3 line 45- What does Western concept of PTSD mean? Be more clear that it is the impact of sexual violence that has not been studied and instead of sequala maybe say burden? It sounds out of place here.

p.3 line 49—remove & , write out “and” ; also can you describe more what you mean by typologies related to culturally reinforced idioms and specifically draw this back to female asylees?

p. 4 lines 26-29 remove the quotes for these constructs i.e., gang violence, domestic abuse – it is not needed

p. 4 line 35—elaborate more on the feminist theories you are using

p. 5 section on Gender Based Violence for Latinas, I think this section should be in the initial framing and come before section Asylum seeking women and gender-based violence.

Something like: “We know studies have demonstrated that women in Central and South America have experienced a disproportionate amount of gender-based violence: domestic and sexual violence (provide your working definitions)” and then incorporate the data from Domestic Violence and Sexual Violence sections next. Then go into Asylum Seeking women & gender based violence section and ending with The Current Study.

Why is Venezuela called out? This does not make sense to me and I would suggest removing it or clarify it and tie it back into the discussion if it is needed.

Methods

p.9 line 31—be consistent with formatting, this is the only description you have n=51 the other two studies you say “a total of”

I would also suggest combining the participants section with the design, it is hard to read with these separated out. That way you have clear descriptions of each study back to back.

You may consider a table to demonstrate the key differences/similarities between studies.

Since Study 1 and 2 were at the same site and had same procedures can you combine the descriptions to avoid repeating yourself?

p.11 lines 45-54- this discussion of the MPP is out of place, can you incorporate into the introduction?

Results

Table 1—these numbers are not the same as the ones described in the methods, Study 1 has n=54 here but in the methods this is different (n=103). Please clarify and if it is not total sample in Table 1 please say what it is in the title of the Table.

Using the word endorsed sexual violence seems off to me, could you say experienced/witnessed? Whatever the original question was in the survey?

p.14- be consistent with capitalization of survey item names, either capitalize or do not.

Table 3: the Ns do not add up in the table, in the first column I get 86 not 97

Figure 3 was not provided for my review

Discussion

Again I would really tie this back to female seeking asylum seekers and the impacts of GBV on their health outcomes here not general statements about migration from Central and SA, so I would delete this first line of this section.

p. 18 line 52—all of a sudden DV is used as an acronym, I would not use it and keep consistent with the rest of the paper

The Trends Per Study section this first paragraph belongs more in the results, instead I would focus more of your discussion on what are the significance of your results in the context of some of the policies you are mentioning (so elaborate on paragraph 2).

Conclusion

The referenced report is a totally new concept that should not be introduced in the conclusion, a conclusion should be more of the summary of the paper/take home messages and then the final recommendation -- I almost wonder if this report can be mentioned in the initial paragraph of the discussion section, showing the significance of your study there. Again the conclusion also should be focused on how you moved the needle forward in reporting on female asylum seekers from Central and SA.

---

## [Reviewer Report]

General observations

Thank you to the authors for their timely submission. This is an interesting and enlightening manuscript that fills in in a gap about what is known about IPV among asylum seeking Latina immigrants. A big concern in this population is the actual prevalence of these events and the willingness to report them, particularly after Latina immigrants are in the US and have uncertain immigration statuses. Overall, there are some questions related to the methods, measures, and discussion.

Methodology

An important methodological question that should be addressed in the manuscript is: how are the experiences of participants in Study 3 comparable to the experiences of participants in the other studies?

Given that part of Study 3 is recruited on the Mexican side of the border, it is possible that part of the sample in Study 3 did not cross the border to the US side. Although this data is about asylum seeking immigrants from Central and South America, it seems possible that the sexual violence and DV experiences of those who stayed on the Mexican side of the border could have been different than those who did not. Given the cross-sectional data presented in this manuscript, it seems that the sample used in Study 3, in comparison with Studies 1 and 2, is not quite the same and therefore introduces questions about the results presented here. The authors could consider presenting results for the sub-group recruited on the US side; or consider presenting the results of Study 3 by recruitment place along the US-Mexico border, and explain in the methods why they are comparable or not, while also including an explanation of the limitations of comparing participants recruited on both sides of the border.

Measures

Given that this manuscript uses different measures across studies, I believe the questions used in each study should be included as part of the manuscript somewhere (maybe in a table). For example, I am familiar with the Harvard Trauma Questionnaire in a few different versions. Some of the versions I have seen do not have a specific question about domestic violence (physical, verbal or emotional). Many versions of the HTQ have questions about beating(s) to the body or physical harm (that includes beating). But those questions do not necessarily specify that the physical violence perpetrated was due to domestic violence (as an example, and maybe less likely but possible, a participant may have been beaten by a different perpetrator that was not a partner or family member along the migration journey). It is important to clarify if there were other measures included in Studies 1 and 3 to complement or adapt HTQ questions.

Discussion

In the discussion about Domestic Violence prevalence for Study 2 there is not much mention of the COVID-19 pandemic as a possible driver of increased DV. Previous studies have found that COVID-19 was associated with increases in domestic violence:

https://journals.sagepub.com/doi/full/10.1177/15248380211038690

https://www.sciencedirect.com/science/article/pii/S004723522100026X#bb0055

Another question about Study 2 that should probably be mentioned as a limitation could be the sample size – given the small sample size (and I understand it is a convenience sample) it is possible that the trends in DV for Study 2 could look more like Study 1 and Study 3 with a larger sample. Although you are not analyzing for significant differences in these experiences across samples (good explanation in the manuscript why you can’t do this), the prevalence of DV in Study 2 requires careful framing in the manuscript and further exploration in subsequent studies.

Figures

Figures 2 and 3 could benefit from including the recruitment period under the study name on the X-axis, so the viewer can keep the timeframe in mind, because you highlight that there were different policies/global events that affected each period of recruitment.

---

## [Reviewer Report]

This paper addresses an extremely important topic and the authors are to be commended for their successful ability to collect data on a very sensitive topic with a hard-to-reach and highly marginalized population.

My main feedback lies with the objective and the methods. Given the underlying three studies are highly heterogeneous - including a diversity of measures, populations, settings, and time periods (including different immigration policy contexts) I do not feel that comparing prevalence across these studies is methodologically appropriate. I do think that substantially reworking the paper to frame these as case studies and to use them to discuss differences in violence experiences in a more purposeful way that builds on their heterogeneity could still be valuable. For example, the authors could leverage these as case examples rather than attempt to present them as a cohesive methodology. I also recommend a table showing each study’s dates, sample size, eligibility, main purpose, setting, and immigration policies in effect during that time.

The introduction would also benefit from some reorganization and streamlining as I found it difficult to follow and longwinded, with substantial repetition.

---

## [Reviewer Report]

The authors have done an excellent job on their revisions, I just have some minor feedback to improve the clarity of the writing.

Abstract

- There is still clarity needed in the revised abstract. In an abstract it should be immediately clear to readers the context you are studying, so what country is being examined (i.e. the US). The sentence you have on p.12 lines 19-26 would be very beneficial in the abstract (starts with the purpose of this study is to…) I would consider paraphrasing something like this in the abstract.

- The 29% statistic is a bit unclear, is that 29% prevalence for women in the U.S. in general please specify the comparison group.

Introduction

- The example of Venezuela is unclear, I would delete this section (page 8, line 30) especially as the example is not illustrating the point that they experience trauma across the migration journey. Authors only highlight pre-migration experiences.

- P. 10 line 8, you do not need to cite Bott et al as you state “In Bott et al” you can just add (2014) after et al.

- P.10 line 33 instead of endorsed by can you say experienced by or reported by? I see you use the verbiage throughout the paper, if you decide to keep it that is okay, but I will just say the connotation does seem a bit off to “endorse” such a traumatic experience, so as much as you can in the writing of the introduction/discussion I would say experienced or reported, as endorsed can sometimes come across as they are “in favor of/supportive of” an experience. I understand that you can endorse a survey item, but I would just be delicate with that language where you can given the context of this particular experience.

Methods

- There are a few instances where you state the mean in a sentence but then repeat it in parentheses. I would just remove the Mean and leave the SD if you already stated the mean in the sentence itself (for example, see p. 13 line 6, p.17 line 23).

- P. 13 line 24, remove the () around the authors and year as that is part of your full sentence. Same for p.14 line 38.

- P. 13 line 51, I would not call it an anonymous project – it is a study (same on p.14 line 24)

- P.14 line 40- just say during 2023 not the 2023 period

- P.15 line 38- only place you mention USD for incentive amount, just keep it consistent and was this a gift card too like the other studies?

- P.16 line 19 I would not break up the name of the survey measure with the reference in the middle, place at the end

Discussion:

- p. 22 line 26 I believe the phrase is to home in on, so to make it easier I would just say “focuses on” as this is also less colloquial.

- p. 23 line 13 – missing punctuation

- p.24 line 45 “Trends per Study” this belongs in the results section it seems out of place in the discussion especially in the middle of it.

- p.25 line 12- I would move this up in the discussion this does not make sense here especially as a natural conclusion occurs on page 24 lines 42-43

Conclusion

- This novel statistics phrase is a bit unclear, perhaps state: this study provides an in-depth visual statistical analyses demonstrating the rates of ....